# One Pot Synthesis of Graphene through Microwave Assisted Liquid Exfoliation of Graphite in Different Solvents

**DOI:** 10.3390/molecules27155027

**Published:** 2022-08-07

**Authors:** Betül Gürünlü, Çiğdem Taşdelen-Yücedağ, Mahmut Bayramoğlu

**Affiliations:** 1Bioengineering Department, Üsküdar University, Altunizade Mah. Üniversite Sok., Üsküdar, Istanbul 34662, Turkey; 2Chemical Engineering Department, Gebze Technical University, Gebze 41400, Turkey

**Keywords:** one pot synthesis, graphene, microwave irradiation, liquid phase exfoliation, graphite, top-down approach

## Abstract

This study presents an easy and quick method for the synthesis of graphene from graphite in a set of solvents, including n-Hexadecane (n-Hexa), dimethylsulfoxide (DMSO), sodium hydroxide (NaOH), 1-octanol (OCTA), perchloric acid (PA), N,N-Dimethylformamide (DMF), ethylene glycol (EG), and ethylene diamine (ED), via microwave (MW) energy. The properties of final products were determined by X-ray diffraction spectroscopy (XRD), Fourier transform infrared spectroscopy (FTIR), ultraviolet-visible (UV-Vis) spectroscopy, and the four-point probe technique. The XRD spectra of most of the MW-assisted graphene products showed peaks at 2θ = 26.5° and 54°. Layer numbers extend from 2 and 25, and the leading comes about were gotten by having two-layered products, named as graphene synthesized in dimethylsulfoxide (G-DMSO), graphene synthesized in ethylene glycol (G-EG), and graphene synthesized in 1-octanol (G-OCTA). G-DMF has the highest electrical conductivity with 22 S/m. The electrical conductivity is higher when the dipole moment of the used solvent is between 2 and 4 Debye (D). The FTIR spectra of most of the MW-assisted graphene products are in line with commercial graphene (CG). The UV-Vis spectra of all MW-assisted graphene products showed a peak at 223 nm referring to characteristic sp2 C=C bonds and 273 nm relating to the n → π * transition of C-O bonds.

## 1. Introduction

Graphene, a monolayer form of graphite, has gathered significant interest in the past few years due to its outstanding properties, such as high electrical conductivity, extraordinary mechanical strength, and large specific surface area [1]. By means of these superior specifications, graphene has a wide range of application areas, such as supercapacitors [2], energy storage/conversion [3], high performance composite materials [4], biosensors [5], drug delivery [6], field emission devices [7], and nanoscale electronic components [8]. From this point of view, it is crucial to develop synthesis methods in view of a high yield of graphene with fewer defects and functional groups [9]. Since the discovery of graphene in 2004, different synthesis methods, including the micromechanical cleavage of graphite [10], chemical vapour deposition (CVD) [11], epitaxial growth on SiC surfaces [12], and chemical reduction of exfoliated graphite oxide [13], have been introduced to the literature. These methods, which require the utilization of expensive equipment, involve complex reduction/oxidation steps and cause low yield in graphene production [14].

Liquid phase exfoliation (LPE) of graphite presents more economical, easier and a scalable alternative for the graphene production in comparison to Hummers’ method [15,16,17]. LPE is a top-down method which eliminates the disadvantages associated with the traditional Hummers’ method, including using heavy and toxic chemicals and successive production steps. Facilitated exfoliation of graphene via microwave (MW) irradiation enables the synthesis of graphene in a controlled shape without the demand for high pressure and temperature [18]. Utilization of MW for the LPE of graphite can enhance interplanar distance and disorder degree of graphite [19,20,21,22]. MW irradiation weakens/breaks the van der Waals bonds between different graphite layers. Therefore, the graphite layers were exfoliated violently and transformed to graphene due to the intercalation of solvent molecules between the layers, as shown in Figure 1 [23].

The solvents having near zero surface tension can penetrate the gap between the graphite layers. This results in an abrupt expansion, generating a force that facilitates exfoliation of graphene in a liquid [24]. Microwave treatment of graphite to weaken interlayer interactions for exfoliation eases the intercalation of solvent molecules into graphite layers [25]. The surface tensions of these solvents chosen for the experiment are around the range 40 and 50 mJ·m^−2^ which is an optimum range for the MW-assisted exfoliation of graphite. Moreover, Pal et al. found that expanded graphite content increased the dielectric properties of the composite with epoxy. The dielectric constant and dielectric loss factor of expanded graphite produced by chemical oxidation were decreased when the microwave treatment was applied to these composites [26]. 

In this study, a simple and rapid method was presented for graphene synthesis and the one-pot synthesis of graphene from graphite through a MW-assisted solvothermal technique was conducted by using various types of low-cost solvents, such as n-Hexadecane (n-Hexa), dimethyl sulfoxide (DMSO), sodium hydroxide (NaOH), 1-octanol (OCTA), perchloric acid (PA), ethylene glycol (EG), and ethylene diamine (ED). Then, the exfoliation capabilities of aforementioned solvents were compared by examining the properties of final products via X-ray diffraction (XRD, Bruker, Billerica, MA, USA) analysis, the four-point probe technique, Fourier transform infrared spectroscopy (FTIR, Perkin Elmer, Hopkinton, MA, USA), ultraviolet-visible (UV-vis, Perkin Elmer, Hopkinton, MA, USA) spectroscopy, Raman spectroscopy (Raman, Renishaw, Wotter-under-Edge, UK), atomic force microscopy (AFM, Q-Scope, Arnhem, The Netherlands) analysis, Brunnauer–Emmett–Teller (BET, Quantachrome, Graz, Austria) analysis, and scanning electron microscopy (SEM, FEI Philips, Hillsboro, OR, USA) analysis.

## 2. Materials and Methods 

### 2.1. Materials

The pristine graphite powder, grade 3061, was purchased from Asbury Graphite Mills, Inc., New Jersey and labelled as commercial graphite (CGr). Commercial graphene (CG) was acquired from Ningbo Yuanshi New Material Technology Co., Ltd., Ningbo, China and used as the reference. N-Hexadecane (Merck, Darmstadt, Germany, 99.5%), dimethyl sulfoxide (Merck, Darmstadt, Germany, 99.9%), sodium hydroxide (J.T. Baker, Leicestershire, UK, 99%), 1-octanol (Merck, Darmstadt, Germany, 99%), perchloric acid (Merck, Darmstadt, Germany, 70–72%), ethylene glycol (ZAG Chemicals, Istanbul, Türkiye, 99.3%), and ethylene diamine (Merck, Darmstadt, Germany, 99%) were of analytical grade.

### 2.2. Experimental System and Synthesis Method

It is worth to be mentioning that the graphene synthesis protocol that is presented in this study is a totally novel method. Firstly, the graphite was dispersed in various solvents at a concentration of 100 mg/mL. The mixture was then sonicated just prior to MW to prevent agglomeration via HD 2200 SonoPuls Homogenizer (Bandelin^®^, Berlin, Germany) under 200 W, 35 kHz, mode 5, and 50% power for 10 min. Afterwards, MW irradiation was applied to the reaction mixture in a 80 mL volume of Teflon vent-and-reseal vessel under the conditions of 180 °C, 150 W for 30 min by a multimode Start-S model (Milestone S.r.l., Sorisole, Italy) MW furnace shown in Figure 2. The MW system includes an infrared (IR) sensor and a fluoroptic (FO) temperature (ATC-FO-300008 type, Zhu Electronic, Italy) sensor. The FO sensor with an accuracy of ±0.2 °C was immersed into the reaction vessel to measure and control the reaction temperature. Moreover, the outer-surface temperature of the vessel was measured and controlled by an IR sensor with the accuracy of ±1 °C. After the reaction, the resulting mixture was centrifuged at 1200 rpm for 30 min. After the removal of black sediment, the supernatant was vacuum filtered through a 0.22-µm Nylon membrane. Then, the obtained product was oven dried overnight at 80 °C. 

### 2.3. Characterization of Synthesized Samples

The graphene samples were characterized by analyzing the thickness, the layer number, and the electrical conductivity of the final products. In order to determine the layer number, X-ray diffraction (XRD) analysis was performed via Rigaku D-Max 2200 Series equipped with Cu-Kα radiation (λ = 1.54 Å) at a scanning rate of 3° per minute. The tube voltage and the current were 40 kV and 40 mA, respectively. The intensity was determined over a 2θ° angular range of 2–90°. Electrical conductivities of synthesized products were measured using a Keithley 2400 Sourcemeter following the manufacturer’s instructions. Firstly, a graphene sample was placed in a cylindrical copper container equipped with a copper cap, thereafter, compressed by a hydraulic press at 50 bars (5 MPa) for 30 min. The electrical resistivities of final products were evaluated by the 4-point probe method. Graphene powder samples were compressed in a copper container with the help of a joiner’s clamp during the electrical conductivity measurements. The conductivity, σ, was then estimated as in Equation (1) shown at below:σ = l/AR(1)
where σ is the conductivity, l is the thickness length of the compacted graphene powder, A refers to the cross-sectional area of the compacted graphene powder, and R is the resistivity of the compacted graphene powder. Raman spectra were recorded in a Renishaw inVia Raman microscope using a 5× optical lens and a 532 nm laser diode with 50 mW as the excitation source. The Fourier transform infrared (FTIR) spectra of samples were measured through Perkin Elmer Spectrum Two equipped with a germanium (Ge) crystal (Pike Gladi ATR Ge-ATR) in the range of 650–4000 cm^−1^. The graphene samples were further characterized via ultraviolet–visible (UV-Vis, UV Perkin Elmer, Lambda 35) spectroscopy. For UV-vis analysis, the dried samples were dispersed in distilled water by agitating via a magnetic stirrer. Afterwards, some of the dispersion was taken into the 10 × 10 mm vial and spectra were recorded at a range of 200–700 nm. The spectrum has an operation range of 200–700 nm. Brunnauer–Emmett–Teller (BET) surface area values of the products were measured by N_2_ physisorption at 77K using a Quantachrome Autosorb Automated Gas Sorption System. The samples were degassed under vacuum at 380 °C for 20 h prior to measurement. The thickness of the samples was determined via a Universal Scanning Probe Microscope (USPM) model Atomic Force Microscopy (AFM) that was built in 250× video microscope, easy change cantilever, a 40 µm × 40 µm × 4 µm scan tube assembly and Acoustic/Vibration Isolation Chamber (AVIC). The microstructures of the synthesized graphene were determined using a scanning electron microscope (FEI Philips XL30 SFEG SEM).

## 3. Results and Discussion

In this study, graphite was converted to graphene using various solvents with the help of MW energy. The MW-assisted LPE technique was applied to a set of graphite-solvent dispersions in order to exfoliate the graphite to graphene. It is possible to measure the affinity of solvent molecules to graphene layers by physical parameters such as dipole moment (μ), dielectric constant (ε), and surface tension (σ). Moreover, the effect of the surface tension, dipole moment, and dielectric constant of the solvents on layer number and electrical conductivity results were examined in order to increase the product quality. The solvents having larger dipole moment, dielectric constant and surface tensions show better solvent properties than those having poor solvent characteristics for a few exceptions [27]. The chemical compositions of solvents and the lateral sizes of graphene layers have an obvious effect on the dispersibility of graphene layers. Thus, many parameters should be evaluated in order to predict the dissolving behaviour of graphene due to its complex mechanism. The graphene dispersions in different solvents are presented in Figure 3.

In the literature, it is understood that N-methyl-pyrrolidone (NMP) and dimethylformamide (DMF) provide better efficiency in regard to the exfoliation of graphite [17,28,29]. The acidic solvents improve the exfoliation mechanism of graphite by breaking the Van der Waals forces and bonds [30]. Moreover, solvents with a high dielectric constant have been proven to build a good effect on the exfoliation mechanism of graphite by functionalization supported via density functional based electronic structure calculations [31]. Interfacial tension is another important factor in LPE of graphene. In order to overcome or decrease this interfacial tension between the graphite layers, it is recommended to study the solvents which have surface tension between 40 and 50 mN/m [17]. For this reason, the solvents which have surface tension that are in and around this range were chosen for the experiments. The parameters evaluating the efficiency of solvents, such as dipole moments, dielectric constants, surface tensions of used solvents, the estimated layer numbers from XRD data obtained via Scherrer equation, and electric conductivity of the graphene products, are given in Table 1.

The thickness of graphene products and CGr was calculated by substituting the results of XRD analyses into Scherrer’s equation, which is denoted as Equation (2) shown below: D002 = Kλ/Bcosθ,(2)
where D002 is the thickness of the crystallite (thickness of graphene), K is a constant dependent on the crystallite shape (0.89), λ is the X-ray wavelength (0.15406 nm), B is the full width at half maximum (FWHM), and θ is the scattering angle [32,33]. The number of graphene layers was calculated by substituting the value of D002 obtained from Scherrer’s equation into the Equation (3): NGP = D002/d002(3)
where NGP is the layer number of graphene products and d002 is the interlayer space for the (002) peak at 2θ = 26.69° [30,34]. As seen in Table 1, the layer number of CGr was calculated as 35, while the layer number of the CG was determined as 2. The graphene products synthesized in n-Hexadecane, Dimethyl sulfoxide, Sodium hydroxide, 1-octanol, perchloric acid, ethylene glycol, ethylene diamine have 25, 2, 14, 2, 5, 2, and 10 layers, respectively, which confirms the obtained graphene structure. Among these products, those with the lowest graphene layers (G-DMSO, G-OCTA, G-PA, G-EG) were chosen for other analyses. Yield values of the chosen samples were also calculated as 12, 5, 13, and 5 for G-DMSO, G-OCTA, G-PA, and G-EG, respectively. These yield values are compatible with the previous literature [9,17,22,35,36].

Moreover, XRD analyses of final products were performed in order to prove the graphene crystal structure. The XRD spectra of the samples were presented in Figure 4 showing two specific peaks at 2θ = 26.69° and 54.92° which are characteristic of graphene and refer to the (002) plane and (004) reflections [37,38]. The intensities of the characteristic peak shown at 2θ = 26.69° of most graphene products are lower than those of CGr. This result indicates the transformation of graphite to graphene, suggesting that the graphitic lattice of CGr changed its periodic arrangement in the z-direction after exfoliation into graphite flakes [39,40,41].

D-spacing values of graphene products were estimated by using Bragg’s equation (Equation (4)), which is shown at below:(d = (nλ)/(2 × sinθ))(4)
where d is the spacing between the diffracting planes in graphene samples, n is diffraction order (taken as 1), λ is the X-ray wavelength (0.15406 nm), and θ is the scattering angle. The results are presented in Table 2.

According to the Table 2, for the first peak that emerged at 26.69°, d-spacing values are 3.33, 3.37, 3.34, 3.35, 3.34, 3.32, 3.35, 3.33, and 3.32 nm for CGr, CG, G-NaOH, G-DMSO, G-n-Hexa, G-OCTA, G-ED, G-EG, and G-PA, respectively. The peak (002) crystal plane of graphene samples gave an average value of 3.34 nm, demonstrating the existence of graphene with few layered structures. Moreover, for the second peak that emerged at 54.92°, d-spacing values are 1.67, 1.68, 1.67, 1.67, 1.68, 1.67, 1.66, 1.68, and 1.67 nm for CGr, CG, G-NaOH, G-DMSO, G-n-Hexa, G-OCTA, G-ED, G-EG, and G-PA, respectively. 

The relation between the surface tension and layer number is presented in Figure 5. The optimum range for surface tension was determined at 45 mN/m for the estimated layer numbers of synthesized graphene products.

The relation between layer numbers and dielectric constant is given in Figure 6. Comparing the layer number of graphene products, it is concluded that as the dielectric constant increased, few-layered graphene was obtained, as seen in Figure 6. The increase at the dielectric constant had a positive impact on obtaining the less layered graphene product. The optimum range of the solvent dielectric constant for synthesizing the less layered product is around 40.

The relation between electrical conductivity and layer number is given in Figure 7. Among all the MW-assisted graphene samples, the highest conductivity was obtained for G-PA with the value 20.6 S/m. Most samples showed the electrical conductivity around 8 S/m.

The structure of the graphene products was also evaluated by FTIR analyses of CG, G-OCTA, G-DMSO, G-NaOH, G-n-Hexa, G-EG, G-ED, and G-PA, as presented in Figure 8.

The FT-IR spectra of G-PA indicates graphene oxide (GO) formation and reveals the presence of various functional groups, including C-O-C and C-O stretching vibrations observed as a very low intensity band at 1083 cm^−1^ due to the remaining carbonyl groups after the sonication and MW irradiation process [42]. The peak in the spectra at 1260 cm^−1^, corresponding to C-O-C bonds, is from remaining epoxides and other ether groups [43]. The peak at 1399 cm^−1^ indicates the residual of solvent coming from filtration process [44]. The sp^2^ C=C bonds was found at 1618 cm^−1^ which have been assigned to skeletal vibrations of graphitic domains [45,46]. The peak at 1740 cm^−1^ is characteristic of -COOH (-C=O in carboxylic acid), which is known to appear after purification steps following reaction and exposure to air [47]. The bands at 2360 cm^−1^ correspond to the asymmetric stretch mode of atmospheric carbon dioxide, which is routinely observed in the background scan on an FTIR measurement, while the peak is caused by the absorption of CO_2_ from the air [48]. The broad peak around 3400 cm^−1^ was due to O-H stretching vibrations [49]. The FTIR spectra of other graphene products showed no sharp peaks and were compatible with the FTIR spectrum of CG.

The UV-vis spectroscopy is one of the most efficient techniques showing the presence of the bonds and the functional groups in the graphene structure. The UV-vis spectra of commercial product (CG) and the MW assisted synthesized graphene products are presented in Figure 9.

The absorbance peak at 223 nm is related to the π → π * electron transition of C-C ring. The absorbance peak at 273 nm is related to n → π * transition of C-O bonds [30]. Previous literature proved that the 280 nm peak is characteristic to graphene [50,51]. The absorbance peak at 240 nm occurred due to the red-shifting of the absorbance peak domain at 223 nm due to the UV-radiation during the measurement [52]. The UV-vis spectra of MW-assisted graphene products are compatible with that of CG. The 237 nm peak is associated with graphene oxide dispersions.

In order to approve the graphene structure, Raman spectroscopy was performed using final graphene samples. The Raman spectra consisting of CG, G-PA, G-EG, G-DMSO, and G-OCTA are given in Figure 10.

The Raman spectra of all graphene products showed a D band at 1350 cm^−1^, G band at 1566 cm^−1^, and 2D band at 2689 cm^−1^. The average value of I_G_/I_2D_ for all samples is 1.72. This indicates that all samples have single layer. A noise peak around 2315 cm^−1^, which is attributed to C-O structure, was seen for all the samples [53]. These results show that the synthesized samples demonstrate graphene structure. In accordance with FTIR results, G-OCTA demonstrated a low graphene structure. In FTIR results, the broad peak shown at 1618 cm^−1^ is characteristic to sp^2^ C=C bonds. In the Raman graph, the D band, also known as the disorder band, occurs due to lattice motion away from the middle of the Brillouin zone and its presence between 1270 and 1450 cm^−1^. Thus, this means that G-OCTA does not demonstrate a good graphene structure.

Brunnauer–Emmett–Teller (BET) surface area values of products labeled as CG, G-OCTA, G-EG, G-DMSO, G-PA, and G-n-Hexa are given in Table 3.

According to Table 3, BET surface area values are 43.554, 19.330, 119.571, 143.395, 157.246, and 78.936 m^2^/g for CG, G-OCTA, G-EG, G-DMSO, G-PA, and G-n-Hexa, respectively. It can be concluded that the graphene products named as G-EG, G-DMSO, and G-PA having the most promising layer number results also have large BET surface area values. CG has a smaller BET surface area than MW-assisted synthesized graphene products. Kamelduski et al. proposed that graphene with a high surface area can also be used in energy storage devices, especially Zn–air batteries [54]. 

The thickness and layer numbers were also estimated by AFM analyses. The AFM results of final graphene products are given in Figure 11.

The thickness of G-DMSO, G-EG, G-OCTA, G-PA, and CG is determined as 2056 nm, 418.85 nm, 1435 nm, 587 nm, and 9.87 nm and the layer numbers of G-DMSO, G-EG, G-OCTA, and CG are calculated as 6137.313, 1250.298, 4283.58, 1752.23, and 29.02, respectively. When comparing with CG, produced graphene samples showed multilayered structures. While comparing the layer numbers obtained by AFM with the layer number values that were calculated by XRD and presented in Table 1, it is understood that AFM values are extremely larger than the XRD ones. It can be explained that it is very difficult in practice to obtain very precise values for the height of graphene, as it can depend upon changing cohesive forces between graphene and supporting substrates. Sometimes, the graphene surface absorbs water vapors, and a very thin layer of water is formed on the graphene surface, while different contaminations can remain on the surface of graphene. Hence, it can be very hard to determine actual thickness of the graphene products by AFM comparing to XRD.

SEM results show that the obtained graphene products have a wrinkled and agglomerated structure. As seen in Figure 12b, the layered structure of graphene was presented clearly. In Figure 12b, rod-shaped structures were also observed. Moreover, most samples showed flat-shaped and folded structures. As can be understood from the SEM image of CG, thin layer and single layer graphene can be seen from the image labeled as Figure 12h [55]. This layered structure can be detected in the most images that were taken from the synthesized samples shown in Figure 12a–g. It can be concluded that multilayered structures decreased to one layer, or few layered structures obtained by means of the MW-assisted solvent-based graphene synthesis. 

## 4. Conclusions

This study presents a quick and easy way for synthesizing graphene from graphite via microwave irradiation. This method depends on LPE of graphite in different solvents whose surface tensions are in the range of 25–75 mN/m. LPE is a top-down method which eliminates the disadvantages related to the traditional Hummers’ method, including using heavy and toxic chemicals and successive production steps. After the reaction and purification steps, obtained carbon products were characterized by using different characterization methods, such as XRD, four-point probe electrical conductivity measurement, FT-IR analysis, UV-vis spectroscopy, Raman spectroscopy, AFM spectroscopy, and SEM analysis techniques. The XRD spectra of the samples showed two peaks at 2θ = 26.5° and 54° which are characteristic of graphene. According to XRD results, obtained samples have 2–25 layers. Most of the products are few layered. Among these products, those with lowest graphene layers were determined as 2, 2, 5, and 2 for G-DMSO, G-OCTA, G-PA, and G-EG, respectively. When the dielectric constant of the used solvent increases, the electrical conductivity values increase too. The FTIR spectra of most of the graphene products showed no sharp peaks and were compatible with the FTIR spectrum of CG. The UV-Vis spectra of all the graphene products display three peaks 223, 240, and 273 nm which refer to the π → π * electron transition to C-C ring, red-shifting of 223 nm domain, and n → π * transition of the C-O bond. The Raman spectra of all products showed peaks at D band at 1350 cm^−1^, G band at 1566 cm^−1^, and 2D band at 2689 cm^−1^ which are characteristic to graphene. AFM analyses demonstrated the multilayered structure of some of the samples. Moreover, SEM results proved that the obtained products have a wrinkled and agglomerated structure as is usually encountered in the graphene images.

## Figures and Tables

**Figure 1 molecules-27-05027-f001:**
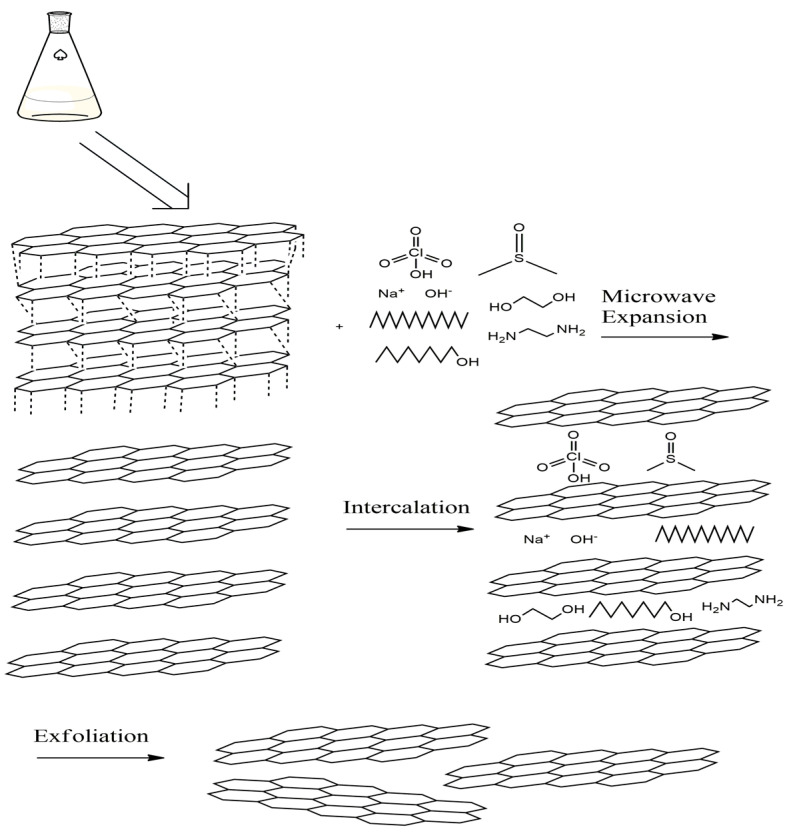
MW-assisted liquid phase exfoliation of graphite to graphene.

**Figure 2 molecules-27-05027-f002:**
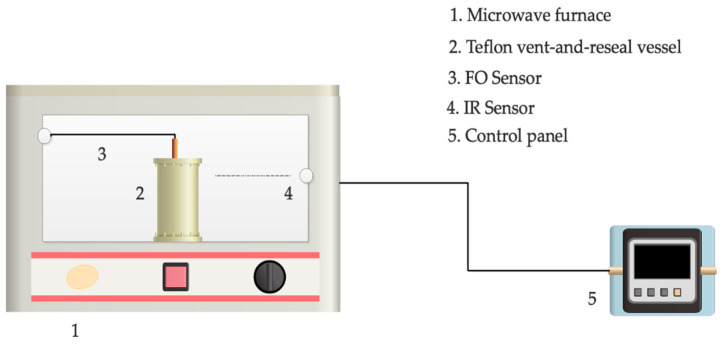
The experimental system with a multimode microwave furnace: Reaction was performed inside a Teflon vent-and-reseal vessel.

**Figure 3 molecules-27-05027-f003:**
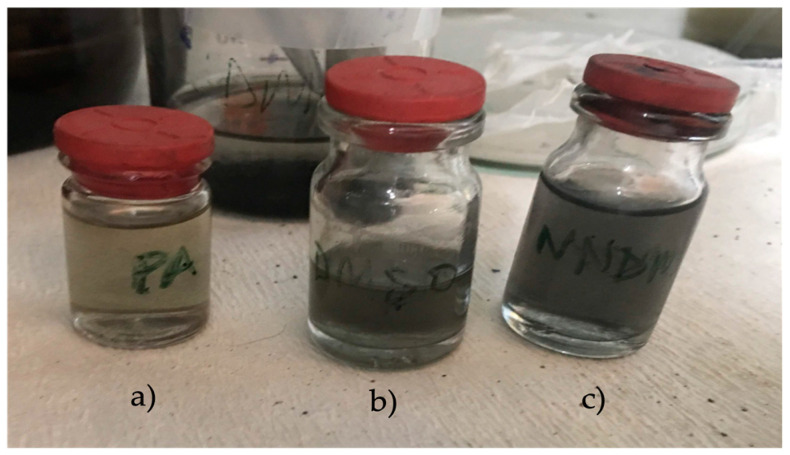
The graphene dispersions in different solvents; (**a**) perchloric acid (PA), (**b**) dimethyl sulfoxide (DMSO), (**c**) N,N-dimethyl formamide (DMF).

**Figure 4 molecules-27-05027-f004:**
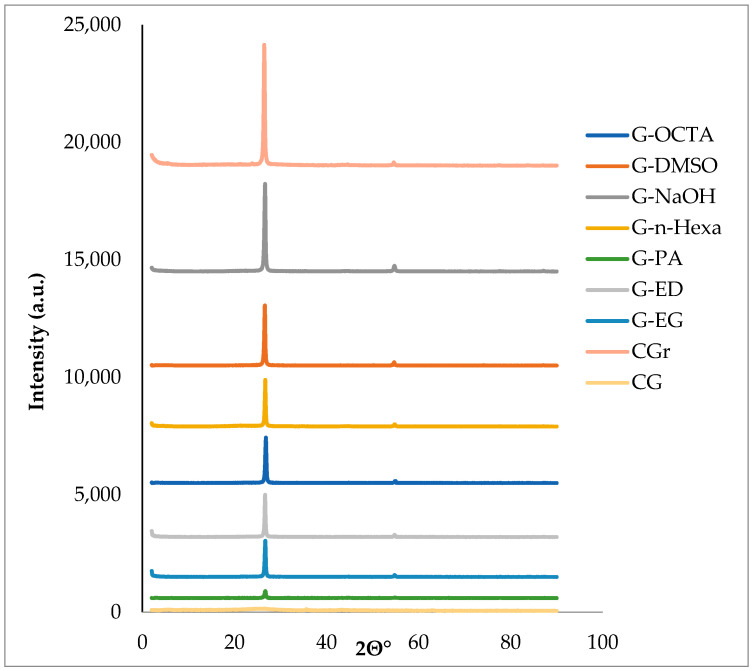
XRD spectra of commercial products and MW-assisted graphene samples.

**Figure 5 molecules-27-05027-f005:**
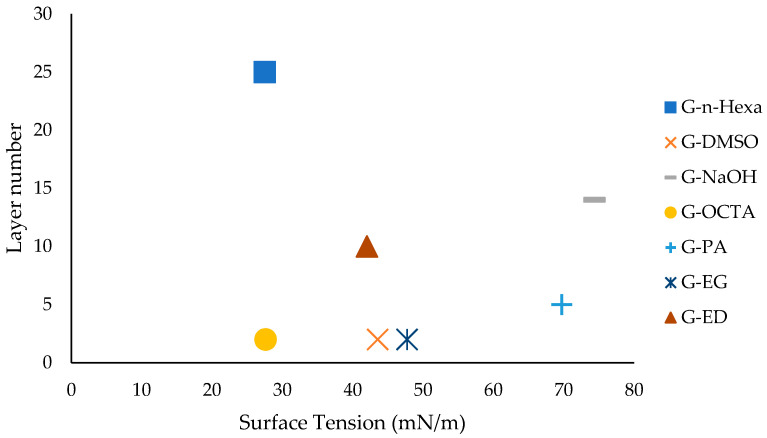
Relation between layer number and surface tension.

**Figure 6 molecules-27-05027-f006:**
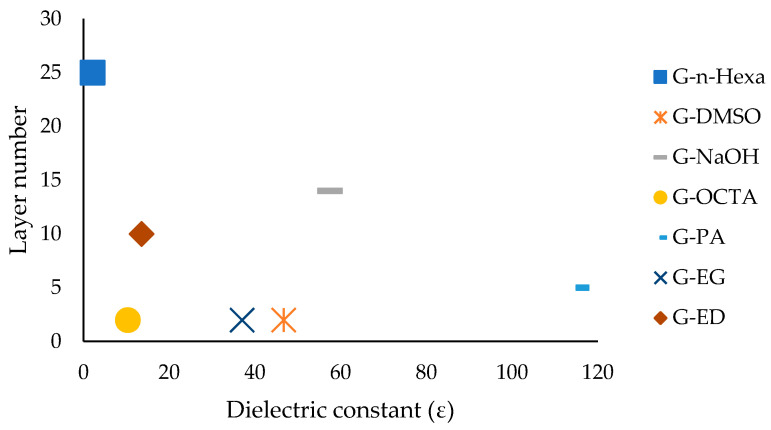
Relation between layer numbers and dielectric constant.

**Figure 7 molecules-27-05027-f007:**
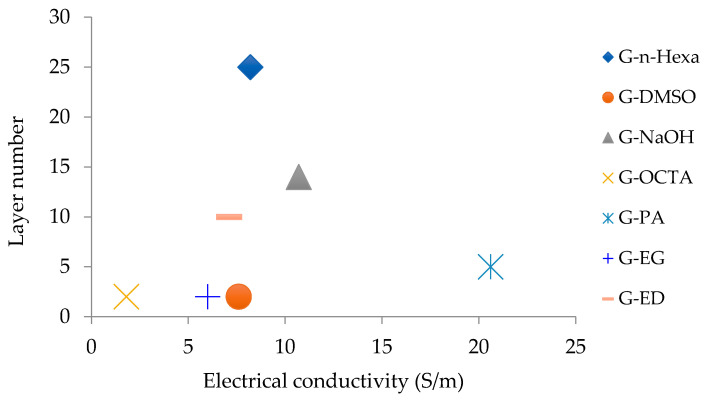
Relation between layer numbers and electrical conductivity.

**Figure 8 molecules-27-05027-f008:**
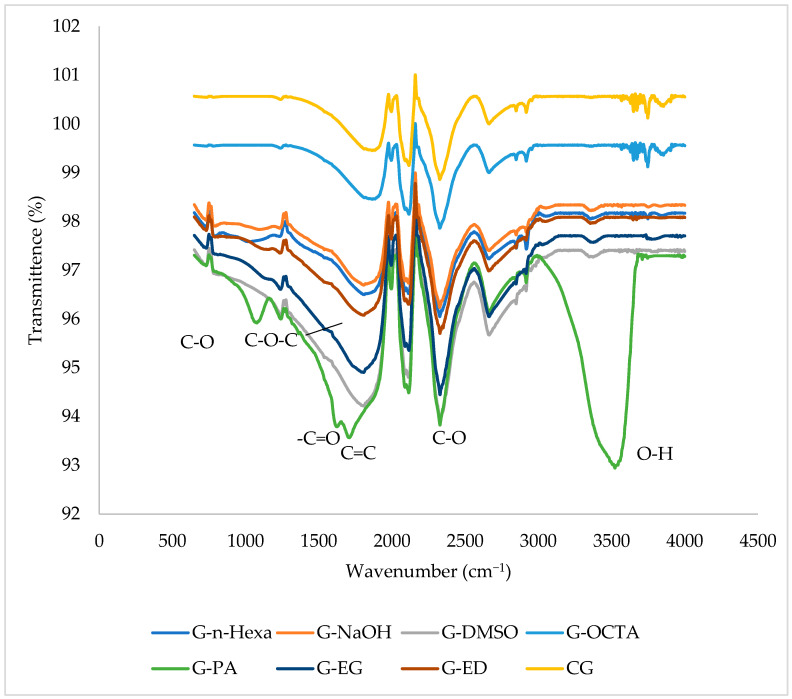
FT-IR spectra of graphene products.

**Figure 9 molecules-27-05027-f009:**
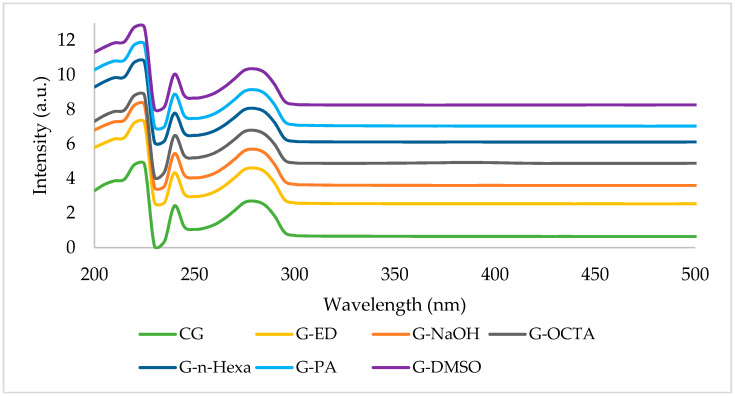
UV-vis spectra of MW-based synthesized graphene products.

**Figure 10 molecules-27-05027-f010:**
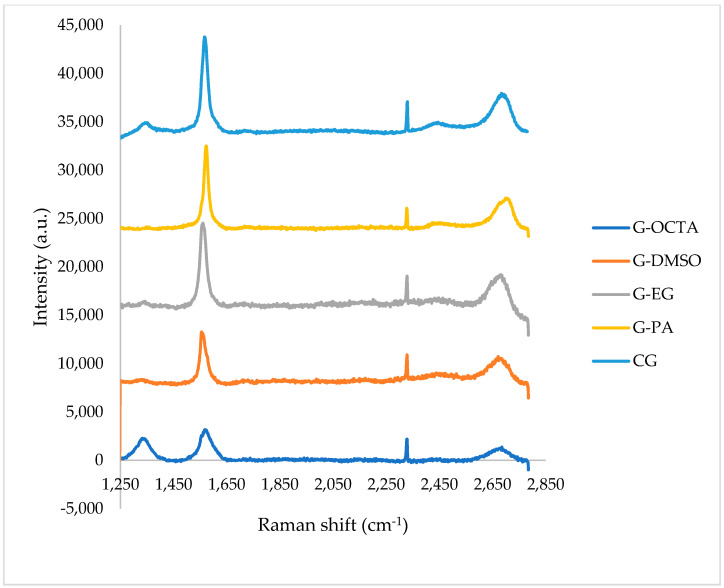
Raman spectra of graphene products labelled as CG, G-OCTA, G-EG, G-DMSO, G-PA.

**Figure 11 molecules-27-05027-f011:**
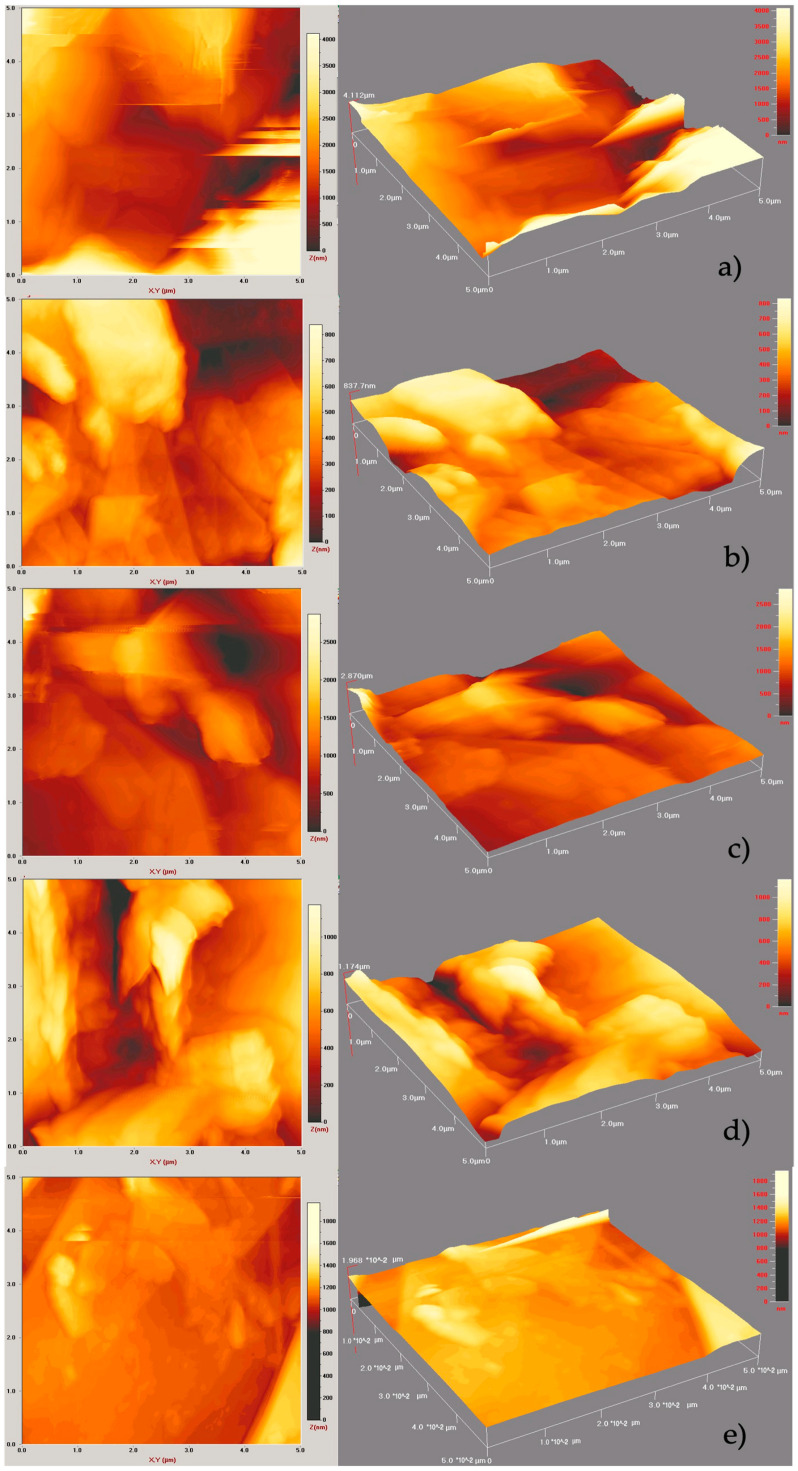
AFM images of MW-assisted synthesized graphene products (**a**) G-DMSO, (**b**) G-EG, (**c**) G-OCTA, (**d**) G-PA, (**e**) CG.

**Figure 12 molecules-27-05027-f012:**
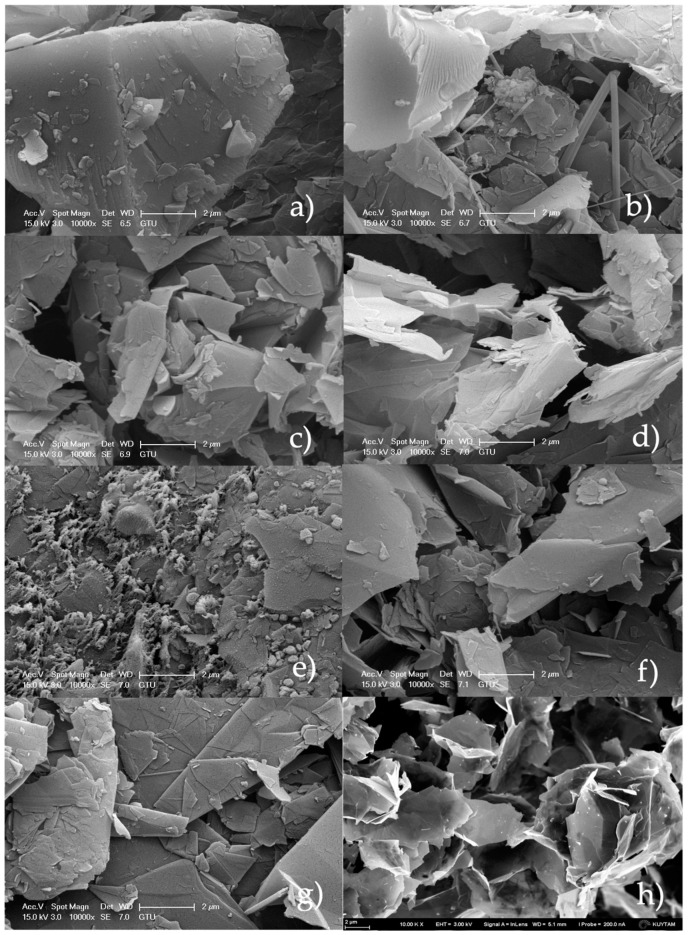
SEM images of MW-assisted synthesized graphene products and CG; (**a**) G-n-Hexa, (**b**) G-NaOH, (**c**) G-DMSO, (**d**) G-OCTA, (**e**) G-PA, (**f**) G-EG, (**g**) G-ED, and (**h**) CG.

**Table 1 molecules-27-05027-t001:** Physical and electrical properties of used solvents.

Solvent Name	Dipole Moment (D)	Dielectric Constant	Surface Tension (mN/m)	The Layer Numbers	Elect. Cond. of Products (S/m)	Yields (wt%)
n-Hexadecane	0.06	2.10	27.47	25	8.20	-
Dimethyl sulfoxide	3.96	46.70	43.54	2	7.60	12
Sodium hydroxide	6.83	57.50	74.35	14	10.70	-
1-octanol	1.68	10.30	27.60	2	1.80	5
Perchloric acid	2.27	115	69.69	5	20.60	13
Ethylene glycol	2.00	37	47.70	2	6.00	5
Ethylene diamine	1.83	13.50	42.00	10	7.10	-
Commercial graphite	n/a	n/a	n/a	35	317	-
Commercial graphene	n/a	n/a	n/a	2	115	-

All the values were measured by the authors.

**Table 2 molecules-27-05027-t002:** d-spacing values of graphene products.

Product Name	2-Theta	Theta	d-Spacing (nm)
CGr	26.74	13.37	3.33
	54.92	27.46	1.67
CG	26.42	13.21	3.37
	54.38	27.19	1.68
G-NaOH	26.68	13.34	3.34
	54.96	27.48	1.67
G-DMSO	26.60	13.30	3.35
	54.86	27.43	1.67
G-n-Hexa	26.66	13.33	3.34
	54.76	27.38	1.68
G-OCTA	26.82	13.41	3.32
	55.00	27.50	1.67
G-ED	26.62	13.31	3.35
	55.46	27.73	1.66
G-EG	26.74	13.37	3.33
	54.76	27.38	1.68
G-PA	26.80	13.40	3.32
	54.84	27.42	1.67

**Table 3 molecules-27-05027-t003:** Brunauer Emmett Teller (BET) surface area values of graphene products.

Product Name	CG	G-OCTA	G-EG	G-DMSO	G-PA	G-n-Hexa
BET Surface Area (m^2^/g)	43.554	19.330	119.571	143.395	157.246	78.936

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
