# Peer review of "One Pot Synthesis of Graphene through Microwave Assisted Liquid Exfoliation of Graphite in Different Solvents"

_molecules, 2022, doi:10.3390/molecules27155027_

Round 1
Reviewer 1 Report
It seems that the authors have improved the quality of their manuscript. One concerning issue is the resolution of some figures (e.g., Figure 1). I recommend the manuscript to be published in Molecules after a minor revision.
Reviewer 2 Report
The present paper represents a revised previous manuscript (ID molecules-1302174) entitled "One pot synthesis of graphene through microwave assisted liquid exfoliation of graphite in different solvents". In the initial manuscript, the Reviewer #2 asked the authors for some revisions. The authors responded to all observations in a cover letter, which is good, but did not operate any changes in the revised manuscript. Without these modifications I can’t recommend the publishing of the revised manuscript (ID molecules-1836279).
Reviewer 3 Report
This article describes the exfoliation of graphite into graphene using the microwave. Many physiochemical characterizations of materials have been shown in the manuscript. The manuscript is structured and written well. However, characterization and many issues should be fixed before publication.
1) A table is needed to explain the sample notations that were used without explanation.
2) The d-spacing of CG is missing.
3) How the layers are calculated should be mentioned clearly?
4) However, the d-spacing of prepared graphene in different solvents does not show much difference when compared CG of CGr?
5) Authors claimed 25 layers, to confirm this TEM images and AFM with height profiles are advised.
6) Figure 7 caption needs correction, Are the spectra for just solvent or composite?
7) Raman spectra for all composite should be included and the presentation of the Raman spectrum needs improvement.
8) The quality of images needs improvement.
9) Why noise peak appears at 2315 cm-1?
10) The authors used seven solvents, so, including CG and CGr, there are nine samples. But some studies showed results from few samples? for ex. Raman, BET, AFM, IR?
11) The layer number from AFM and XRD are not the same?
12) SEM images presented here fail to explain the exfoliation of graphite into graphene.
13) There are many reports explaining the graphene dispersion or exfoliation in different solvents that should be compared with present studies.
14) Photographic images of graphene dispersion in the different solvent is advised.
Reviewer 4 Report
Manuscript presented by Betül Gürünlü et al. shows a study about graphene synthesis in one pot via microwave energy. An already well written and prepared manuscript. Easy to read and follow. Some aspects should be improved (or need to be re-written).
I recommend the article to publish but first the paper should be improve. My decision – reconsider after minor revision. Comments to be considered, in order to further improve the manuscript quality:
(1) Authors wrote: "Also, it is discovered that the dielectric constant and loss factor of expanded graphite produced by chemical oxidation is raised when the composites are synthesized by microwave curing [26]". What does it mean? Please elaborate topic in one sentence.
(2) Please add information about the novelty of your work. What progress against the most recent state-of-the-art similar studies was made?
(3) It is unclear, the graphene synthesis protocol used in this work is a standard procedure. If yes - please add the appropriate reference, if no - clearly mark this fact in manuscript.
(4) In order to show this manuscript is suitable for the publication in presented journal please include into manuscript some Molecules publication.
(5) Add “One pot synthesis” and “graphene” to keywords.
(6) The style of reference should be improve (see template).
(7) The English correction is necessary.
Round 2
Reviewer 2 Report
The authors improved their manuscript and responded properly at all reviewer's suggestions.
This manuscript is a resubmission of an earlier submission. The following is a list of the peer review reports and author responses from that submission.
Round 1
Reviewer 1 Report
In their manuscript “One pot synthesis of graphene through microwave assisted liquid exfoliation of graphite in different solvents” Gürünlü et al.investigated the exfoliation of graphene from graphite in solvents via use of microwave irradiation.
The manuscript is not in a good shape and I believe it is not publishable in its current state. I will not support accepting the manuscript for publication in Molecules.
I outlined the suggestions and questions below for improving the clarity of the manuscript.
Lots of typos and grammatical errors (even in Abstract): Lines 19, 40, 60, 177, 212, 219, 238, 255
Introduction should be re-written as it does not convey what is discussed in the manuscript.
All the figure and table captions need to be elaborated.
It should be mentioned why the solvents used in the manuscript were chosen.
“The layer numbers” does not mean the same “the number of layers”. Please correct accordingly.
How many experiments did the authors do per system? Can they provide any error bars?
The authors provided many figures but only a few layers of text for most of them.
Figures are not standardised.
cm-1 --> cm-1 (in many instances)
Figures 10 and 11 are not readable.
Author Contributions section should be taken seriously.
Page 2 Lines 28-29: Provide example values for electrical conductivity, extraordinary mechanical strength and large specific surface area.
Page 2 Lines 30-31: Supercapacitors are energy storage devices, too. Why did you mention it separately?
Page 2 Lines 33-34: The authors say “it is crucial to develop synthesis methods in high yield of graphene with less defects and less functional groups”. But they did not provide the reason of why?
Page 2 Line 43: Can you give some examples to toxic chemical used (e.g. X and Y)?
Page 5 Line 141: Why do authors mean by “the product quality”?
Page 7 Lines 199-201: The authors say “The peak (002) crystal plane of graphene 199 samples gave an average value of 3.34 Ã…, demonstrating the existence of graphene with 200 few-layered structures.”. This need to be elaborated.
Page 7 Lines 205-206: The authors say “The optimum range for surface tension was determined at 45 mN/m for the estimated layer numbers of synthesized graphene products.” This is not agreeable with Figure 4.
Page 8 Lines 209-211: same problem.
Page 9 Line 243: The authors say “while the peak caused by the absorption of CO2 from the air”. This is not relevant.
Page 11 Lines 279-281: There are four sample mentioned but five numbers were provided.
Reviewer 2 Report
The authors have studied a quick and easy way for synthesizing graphene from graphite using various solvents (n-Hexadecane, dimethylsulfoxide, sodium hydroxide, 1-octanol, perchloric acid, N,N-Dimethylformamide, ethylene glycol and ethylene diamine) via microwave irradiation. After the reaction and purification steps, the prepared products were characterized by XRD, 4-point probe electrical conductivity measurement, FT-IR analysis, UV-vis spectroscopy, Raman spectroscopy, AFM spectroscopy and SEM analysis techniques. The results revealed that depending on physic-chemical properties of solvents could be prepared graphene from graphite via microwave irradiation with layer numbers range from 2 and 25.
Some revisions are necessary:
1. At page 1, lines 18-19: The phrase "...graphene synthesized in 1-octanol (G-OCTA) with 2." seems to be unfinished. Please reformulate it.
2. At page 3, line 89: It is about Figure 2 (not 1). Please correct.
3. At page 5, line 161 (Table 1): The values from Table 1 were measured by the authors? If yes, please specify that. If not, please insert a new column with references for each solvent.
4. At page 6, line 162: Please define the abbreviations at first use (e.g. CGr, GO at line 232).
5. At pages 6-7, lines 163, 168, 191: For better view/ understanding, I suggest to the authors to write the equations on separate lines, as in "Molecules" journal template.
6. At page 8, line 225: "...evaluated by FTIR analyses of CG, G-OCTA, G-DMSO, G-NaOH, G-n-Hexa, G-ED, G-PA presented in Figure 7". Looking at Fig.7, seems that EG should be replaced with CG (or vice-versa: replace CG with G-EG in text, at line 225). In the second case, please insert also the CG spectrum in Fig.7.
7. At page 9, line 229 (Figure 7): The samples labels should have a "G" in front (e.g. G-n-Hexa, G-NaOH, G-DMSO, etc.).
8. At page 9, line 230: It is about Figure 7 (not 6). Please correct.
9. At page 9, line 251 (Figure 8): The spectra of G-n-Hexa and G-PA are overlapped. Please separate them.
10. At page 10, line 264 (Figure 9): The CG (the reference) spectrum is missing. Why? Instead, it appears the G-EG spectrum, which is not presented in previous two figures. Why?
11. At page 10, line 264 (Figure 9): Please explain why the D band at 1350 cm-1 of G-OCTA is much higher than the others.
12. At page 11, line 275 (Figure 10): Please enlarge the font size of the labels (a, b, c, d) to be more visible.
13. At page 11, line 275 (Figure 10): It would be useful to add also the AFM analysis of the reference (CG) in Fig.10.
14. At page 11, lines 279-281: The authors should provide (comment) a comparison of these results with calculated values from Table 1 and offer some explanations.
15. At page 12, line 283 (Figure 11): It would be useful to insert also the SEM image of the reference (CG). Also, please enlarge the font size of the labels (a, b, c... g) to be more visible.
Reviewer 3 Report
In this manuscript, the authors present an easy and quick way for synthesis of graphene from graphite in various solvents, and then exhibit the corresponding material analysis data. However, the achievement of their products in this work lacks substantial impact on the material research field. Also, this work falls short in providing technical innovation for preparing graphene. This work looks like a routine work, and all the data are presented in poor quality (see the Figures of this manuscript). For these reasons, I cannot recommend this manuscript to be published in the Journal, Molecules, with an Impact Factor of 4.411.
Nevertheless, I would still like to raise some questions about the present work to bring the authors’ attention.
- In the abstract, they mentioned below sentence:
“The electrical conductivity is higher when the dipole moment of the used solvent is between 2 - 4 Debye (D).”
This is not totally exact description; in Table 1, just compare the case of “n-Hexadecane” to “Ethylene glycol” and “Dimethyl sulfoxide”.
- In the introduction section, they mentioned below sentence:
“By means of these superior specifications, graphene has a wide range of application area such as supercapacitors [2], energy storage/conversion [3], high performance composite materials……….”
Actually, supercapacitors have been included in energy storage. Besides, what is high performance composite materials?
- In the introduction section, they mentioned below sentence:
“From this point of view, it is crucial to develop synthesis methods in high yield of graphene with less defects and less functional groups”
Obviously, this work didn’t reach this requirement. Their products is synthesized in low yield and possess defects and plenty of functional groups.
Moreover, their products show much lower electrical conductivity than that of commercial graphene (see Table 1).
- In the introduction section, they mentioned below sentence:
“The surface tensions of these solvents chosen for the experiment are around the range 40 and 50 mJ·m−2 which is an optimum range for the MW-assisted exfoliation of graphite.”
The related references should be cited here.
- Their products are graphene oxide rather than graphene. So, their tittle should be revised to be “One pot synthesis of graphene oxide through microwave assisted…….”.
- In Table 1, why several yields didn’t be provided?
- In paragraphs of Raman analysis, they mentioned below sentence:
“The average value of IG/I2D for all samples is 1.72. This indicates that all samples have single layer.”
This result seems didn’t match the results obtained from the data of XRD, AFM and SEM.
“A noise peak around 2315 cm-1 was seen for all the samples.”
They should try to explain this.
- In paragraphs of AFM analysis, they mentioned below sentence:
“The thickness of G-DMSO, G-EG, G-OCTA, G-PA is 2056 nm, 418.85 nm, 1435 nm, 587 nm and the layer number of G-DMSO, G-EG, G-OCTA, and G-PA is 2937, 6137.313, 1250.298, 4283.58, 1752.23, respectively.”
According to these layer numbers, how can they claim the products as graphene rather than graphite?
- In paragraphs of SEM analysis, they mentioned below sentence:
“In Figure 11b, also rod-shaped structures were observed.”
They should try to investigate and explain what is the rod-shaped product?
A final suggestion goes to the English writing and incorrect grammar in this manuscript. An English editing to revise the manuscript is recommended. Few examples obtained from the manuscript are illustrated below:
(i) Layer numbers range from 2 and 25, and the best results were obtained by graphene synthesized in dimethylsulfoxide (G-DMSO), graphene synthesized in ethylene glycol (G-EG), graphene synthesized in 1-octanol (G-OCTA) with 2. G-DMF has the highest electrical conductivity with 22 S/m.
(ii) Microwave treatment of graphite to weaken interlayer interactions for exfoliation eases the intercalation of solvent molecules into graphite layers.